# Behavioral, Emotional and Social Apathy in Alcohol-Related Cognitive Disorders

**DOI:** 10.3390/jcm10112447

**Published:** 2021-05-31

**Authors:** Maud E. G. van Dorst, Yvonne C. M. Rensen, Masud Husain, Roy P. C. Kessels

**Affiliations:** 1Centre of Excellence for Korsakoff and Alcohol-Related Cognitive Disorders, Vincent van Gogh Institute for Psychiatry, 5803 DN Venray, The Netherlands; yrensen@vvgi.nl (Y.C.M.R.); roy.kessels@donders.ru.nl (R.P.C.K.); 2Donders Institute for Brain, Cognition and Behaviour, Radboud University, 6525 XZ Nijmegen, The Netherlands; 3Nuffield Department of Clinical Neurosciences, University of Oxford, Oxford OX1 2JD, UK; masud.husain@ndcn.ox.ac.uk; 4Department of Experimental Psychology, University of Oxford, Oxford OX1 2JD, UK; 5Tactus Addiction Care, 7418 ET Deventer, The Netherlands; 6Department of Medical Psychology, Radboud University Medical Center, 6525 GA Nijmegen, The Netherlands

**Keywords:** apathy, neuropsychology, Korsakoff’s syndrome, alcohol use disorder, alcohol-related cognitive impairments

## Abstract

Apathy is a fundamental neuropsychiatric symptom of Korsakoff’s syndrome (KS) and has also been reported in patients with alcohol use disorder with no (AUD) or less severe cognitive impairments (ARCI). However, research on the nature of apathy is limited in these groups. Aim of this study was to examine the multidimensional nature of apathy in patients with KS, ARCI and AUD. Moreover, we examined differences between apathy ratings by patients and their professional caregivers, and related apathy to everyday functioning and overall cognition. Twenty-five patients with KS, 25 patients with ARCI and 23 patients with AUD participated in this study. Apathy was measured using the apathy motivation index (AMI), which distinguishes behavioral, emotional and social apathy. Both patients and professional caregivers reported social apathy as the most prominent symptom, compared to behavioral and emotional apathy. Apathy ratings did not differ across the three patient groups. Discrepancies between patient and caregiver ratings were observed in patients with KS and ARCI, with more severe apathy reported by caregivers. Caregiver-reported behavioral and social, but not emotional, apathy was related to everyday functioning. These results show that apathy is present in a substantial proportion of patients with alcohol addiction with or without cognitive impairments.

## 1. Introduction

Chronic and excessive alcohol consumption negatively affects brain functioning [1,2,3] and may result in various degrees of cognitive impairment—in attention, memory, visuospatial functions and executive functions—in patients with alcohol use disorder (AUD). However, abstinence improves or optimizes patients’ cognitive functioning [4]. After a period of abstinence, some patients with AUD do not show any cognitive deficits (in the current study referred to as patients with alcohol use disorder, without cognitive impairments or AUD), but others continue to show cognitive impairments even after longer periods of abstinence (in the current study referred to as patients with alcohol-related cognitive impairments, ARCI). Korsakoff’s syndrome (KS) is the most severe neuropsychiatric disorder that may develop in the context of chronic and excessive alcohol consumption. It results from thiamine depletion, typically in the context of chronic excessive alcohol consumption and malnutrition. The syndrome is characterized by memory impairments. In addition, neuropsychiatric symptoms such as apathy, lack of illness insight, flattened affect and confabulations are also present [5]. Although apathy is recognized as a fundamental symptom of Korsakoff’s syndrome, it is poorly studied to date. Moreover, apathy has received limited attention in patients with AUD and ARCI as well.

The definition of apathy has been refined over the last decades. In 1991, Marin [6] formulated the first operational definition and defined apathy as “a lack of motivation that is not attributable to intellectual impairment, emotional distress, or diminished level of consciousness” [6] (p. 245). He postulated three clinical features: (i) reduced goal-directed behavior, (ii) reduced goal-directed cognition and (iii) reduction of emotion that is associated with goal-directed behavior. Starkstein et al. [7] later expanded the definition by stating that the construct of apathy is independent from cognitive impairment or a depressive disorder, so that both these conditions and apathy can coexist within the same person. Recently, the multidimensional nature of apathy was reconsidered, as “behavior” and “cognition” were found to be frequently associated with one another, thus representing one dimension. In addition to a reduction in emotion, impaired social interaction was identified as a separate dimension [8,9]. Accordingly, Robert et al. [10] defined apathy as “a quantitative reduction of goal-directed activity either in behavioral, emotional or social dimensions in comparison to the patient’s previous level of functioning in these areas” [10] (p. 73), as reported by the patient himself/herself, or by others.

With respect to the underlying neural and cognitive mechanisms of apathy, findings to date are inconclusive. At a cognitive level, it has been argued that executive dysfunction is associated with apathy, but findings are inconsistent across studies and patient groups [11]. With respect to the neural correlates, prefrontal-basal ganglia circuits, including the anterior cingulate cortex and striatum have been suggested to underlie apathy, but it is to date unclear how the distinct subtypes of apathy map to specific neural correlates [11]. The few studies on apathy conducted to date in patients with AUD and KS suggest that apathy is present in both groups. To the authors’ knowledge, apathy has not yet been studied in patients with ARCI who do not meet the criteria for KS. One investigation examining patients with substance dependence, including patients with AUD, reported higher degrees of apathy (assessed using the frontal systems behavior scale) [12]. In a report by Yang et al. [13], ratings of apathy (assessed using the Lille apathy rating scale-informant version, LARS-I) were higher for patients with AUD compared to healthy controls. Furthermore, Yang et al. [13] examined the neural basis of apathy in AUD and reported significant decreases in overall cortical thickness in the occipito-temporal cortex (OTC), bilateral superior parietal cortex (SPC) and bilateral inferior parietal cortex (IPC) in patients with AUD compared to healthy controls. The results also showed a negative correlation between the cortical thickness of the OTC, SPC and IPC and apathy ratings. In patients with KS, no studies on the neural correlates of apathy have been performed, but caregivers rated apathy (assessed using informant questionnaires) as the most severe neuropsychiatric symptom of KS [14]. Furthermore, apathy was found to be a persistent symptom in patients with KS, which—unlike psychotic and affective symptoms and agitation/aggression—was not reduced after a cognitive rehabilitation intervention [15].

Gaining an accurate understanding of apathy in patients with AUD, ARCI and KS is complex, due to the limited number of previous studies, investigations examining only one of these patient groups and the variety of instruments used to quantify apathy. It is unclear, for example, whether apathy in patients with alcohol-related cognitive disorders represents a continuous spectrum, with the most severe apathy being present in patients with KS, and less severe apathy in patients with ARCI and AUD respectively. Moreover, previous studies addressed apathy in patients with alcohol-related cognitive disorders as a unitary construct, and some instruments used to examine apathy contain only one “general apathy” item [12,13,14,15], There is increasing evidence, however, that apathy has several components.

Kessels et al. [16] recently examined the multidimensional nature of apathy in patients with KS, using the informant-version of the apathy evaluation scale (AES-I). They found that 72.1% of the patients with KS could be classified as having an apathy syndrome, with the cognitive dimension of apathy evaluated as more severe compared to the emotional dimension of apathy. Furthermore, AES-I scores were found to correlate with overall cognitive dysfunction as measured with the Montreal cognitive assessment (MoCA) and everyday executive dysfunction [16]. In another recent study [17], higher rating of general apathy (assessed using the AES-I total score) were found in patients with KS with and without cerebrovascular comorbidity, as compared to healthy controls, but no significant correlation with executive functioning was found, possibly because of the small sample size. More importantly, the AES-I does not fully cover the most current definition of apathy. The apathy motivation index (AMI) was recently developed as a tool that might better capture the multidimensional nature of apathy, including social apathy, but has not yet been applied in individuals with alcohol-related cognitive disorder.

The aim of the present study is to examine the multidimensional nature of apathy in patients with AUD, ARCI and KS using the recently developed AMI [8]. With the AMI, three dimensions of apathy can be distinguished: behavioral activation (e.g., “I get things done when they need to be done, without requiring reminders from others”), emotional sensitivity (e.g., “I feel awful if I say something insensitive”) and social motivation (e.g., “I start conversations without being prompted”). In addition, we aim to relate the three dimensions of apathy to everyday functioning and overall cognitive performance. Based on previous results [8,9,13,16,17], we hypothesize that all three dimensions of apathy will be present in patients with AUD, ARCI and KS. We expect behavioral apathy to be the most frequently reported dimension of apathy in all three groups and hypothesize that especially behavioral apathy is negatively related to everyday functioning and cognitive performance. Moreover, we examined differences between patient and professional caregiver ratings on the AMI, as impaired awareness of deficits is highly common in alcohol related cognitive disorders and impaired awareness of deficits and apathy are found to be correlated with each other [8,18,19]. We hypothesize that professional caregivers will report higher levels of apathy than patients with KS and ARCI.

## 2. Materials and Methods

### 2.1. Participants

Patients were recruited from the diagnostic clinic, the long-term care unit or the ambulatory support facility of the Centre of Excellence for Korsakoff and Alcohol-Related Cognitive Disorders of Vincent van Gogh Institute for Psychiatry in Venray, the Netherlands. All were evaluated extensively using neurological, psychiatric, neuroradiological and neuropsychological examinations as part of their diagnostic trajectory at the center. The neuropsychological assessment covered the domains of attention, memory, executive functioning, visuospatial functions and social cognition and scores were interpreted by experienced clinical neuropsychologists using all available age-, education- and/or gender-adjusted Dutch normative data and/or available cut-off scores. Test scores were classified as impaired if a score was more than 2 SDs below the normative mean. A neuropsychological disorder was diagnosed if two test scores within a cognitive domain were classified as impaired. The patients themselves, their family and/or professional caregivers and medical records provided background information (including drinking history). The eventual diagnoses were established by a multidisciplinary expert team. All patients were abstinent from alcohol at the time of testing (median abstinence duration 64 days, range from 38 to 19 years), verified by regular urinalysis. None had any evidence of brain pathology unrelated to their alcohol use that would account for their memory deficit (e.g., stroke, tumor or a neurodegenerative disease) or met the criteria for alcohol-related dementia [20]. All participants were screened by a consulting psychiatrist as part of their diagnostic trajectory at the center and none met the DSM-5 criteria for a major depressive disorder [21].

A total of 74 patients were included in this study. Group characteristics are presented in Table 1. The sample consisted of three groups. Twenty-three patients were diagnosed with severe alcohol use disorder in accordance with the DSM-5 [21], but extensive neuropsychological testing did not show disorders in any of the cognitive domains (AUD). Twenty-five patients with alcohol-related cognitive impairments (ARCIs) participated in this study. These individuals met the criteria for DSM-5 alcohol-induced neurocognitive disorder [21] and showed impairments in one or more cognitive domains. Twenty-five patients with Korsakoff’s syndrome (KS) participated in this study. They met the criteria for DSM-5 alcohol-induced major neurocognitive disorder, amnestic-confabulatory type [21]. In addition, the criteria for alcoholic Korsakoff’s syndrome [22,23] had to be met: “a largely irreversible residual syndrome, caused by severe thiamine deficiency and occurring after incomplete recovery from a Wernicke’s encephalopathy, predominantly in the context of alcohol abuse and malnutrition, characterized by an abnormal mental state in which episodic memory is affected out of all proportion to other cognitive functions in an otherwise alert and responsive patient” [23] (p. 10).

The level of formal education was assessed using a scale with seven categories based on the Dutch educational system, 1 being the lowest (less than primary education; i.e., six or less years of education) and 7 the highest (academic degree). Premorbid IQ was estimated using the Dutch version of the National Adult Reading Test [24], Full-scale IQ was measured using the Wechsler Adult Intelligence Scale, Fourth Edition (WAIS-IV) [25]. Between-group differences in demographic and background variables were explored using analysis of variance (ANOVA) for the normally distributed variables, Chi-squared test for sex distribution and Kruskal–Wallis test for educational level and abstinence duration (as the latter variable was skewed due to the inclusion of a few patients from the long-term care unit). The groups were found to be comparable with respect to age, sex distribution, estimated premorbid IQ, Full-Scale IQ scores, and educational level. The sample consisted of more men than women, reflecting the prevalence of alcohol use disorder in the general population. Additionally, the sample consists of individuals with lower estimated premorbid intelligence (with 1 KS-patient and 1 ARCI-patient with an estimated premorbid IQ < 70) as this was not an exclusion criterion. This sample is, however, highly representative of patients in a specialized psychiatric hospital, with similar IQ ranges as in a previous larger sample of these three diagnostic groups [26].

### 2.2. Procedure

The apathy motivation index (AMI) [8] was used to examine apathy. The English-language version of the AMI was translated into Dutch by the second author, checked for inconsistencies and phrasing by the last author. Moreover, the patient-report version of the AMI was transformed to an informant version, by changing all “Is” to “he/shes” (e.g., “I feel sad or upset when I hear bad news” was transformed to “He/She feels sad or upset when he/she hears bad news”). Two expert researchers with knowledge of the field pointed out inadequate expressions and discrepancies in the translated instrument. An independent bilingual translator, with knowledge of the field, translated the instrument back to English. A comparison between the original instrument and the translated version showed no important differences in content. The informant version was also checked against the recently developed English-language informant version [27].

The AMI is an 18-item scale that measures apathy as a multidimensional construct. The three dimensions as distinguished by the AMI are labelled as behavioral activation, social motivation and emotional sensitivity, and are based on factor analysis on both the patient-reported version [8] and the caregiver-rated version [27]. Both versions also showed a good external validity, as they correlated with the Lille apathy rating scale (LARS) patient- and caregiver-reported versions [27,28]. The AMI was completed by patient and professional caregivers during the period from July 2018 to February 2021. Professional caregivers were trained nurses who had known the patient well for at least six weeks. Answers were based on the patient’s behavior in the last two weeks. Each item was rated on a 5-point Likert scale and scored from 0 with higher scores indicating more apathy. We computed the individual average AMI subscale scores as used in previous research and on which the cut-off scores are based [8,27,28], but also computed the individual total AMI subscale scores for descriptive purposes (see Table A1). The scores on the three subscales were used to answer the current research question. The patient-report version of the AMI was completed by 71 out of the total of 73 participants. The informant-version of the AMI was completed by professional caregivers for 62 participants. Both versions of the AMI were completed for a group of 60 patients.

The Dutch version of the Montreal Cognitive Assessment (MoCA) [29] was used to quantify general cognitive functioning. The items of the MoCA add up to a total score with a maximum of 30 with a higher score indicating better cognitive functioning. The MoCA Total score is found to be a valid and reliable global measure for cognitive function and the MoCA memory index score has a good predictive value for memory impairment [24].

The patient competency rating scale (PCRS) [30], completed by professional caregivers, was used to examine everyday functioning. The PCRS is a 30-item scale that measures everyday functioning on a variety of tasks and functions, including activities of daily living, interpersonal functioning, everyday cognitive functioning and emotional functioning. Each item was rated on a 5-point Likert scale with higher scores indicating better competences. We used the PCRS total score that was found to have adequate psychometric properties when used in individuals with cognitive impairment and brain dysfunction [31].

### 2.3. Statistical Analysis

The levels of behavioral, emotional and social apathy were expressed in mean scores for each completed patient-report and caregiver-report version of the AMI. The multidimensional nature of apathy for the current study population was examined using a mixed ANOVA with apathy dimension as within-subject factor (behavioral, emotional and social) and patient group as between-subject factor (AUD, ARCI and KS). This mixed ANOVA was performed for patient-reported and caregiver-reported apathy separately. Differences between patient and caregiver ratings were examined using paired-sample *t*-tests. For each patient group (AUD, ARCI and KS), paired *t*-tests were performed with the patient-reported mean rating and the caregiver-reported mean rating as dependent means for behavioral, emotional and social apathy separately. Pearson correlations were calculated to examine the relationship between behavioral, emotional and social apathy, everyday functioning and cognitive performance.

## 3. Results

Figure 1 represents the distribution of individual ratings of behavioral, emotional and social apathy on the AMI (mean ratings and standard deviations for the three patient groups separately are presented in Table A1 in Appendix A, both for the average and total scores of the AMI subscales). A main effect of dimension was found for both patient-reported and caregiver-reported apathy (*F*(2,67) = 17.41, *p* < 0.001, η_p_^2^ = 0.34; *F*(2,58) = 16.58, *p* < 0.001, η_p_^2^ = 0.36). Both patients and caregivers reported social apathy as more severe compared to behavioral (patients: *F*(1,68) = 33.39, *p* < 0.001, η_p_^2^ = 0.33; caregivers: *F*(1,59) = 14.19, *p* < 0.001, η_p_^2^ = 0.19;) and emotional apathy (patients: *F*(1,68) = 9.69, *p* = 0.003, η_p_^2^ = 0.13; caregivers: *F*(1,59) = 29.88, *p* < 0.001, η_p_^2^ = 0.34).

Overall, patient-reported and caregiver-reported apathy did not differ across the three patient groups (patients: *F*(2,68) = 2.34, *p* = 0.104; caregivers: *F*(2,59) = 0.43, *p* = 0.651). Furthermore, there was no interaction between dimension and group for patient-reported or caregiver-reported apathy (*F*(4,134) = 1.16, *p* = 0.330; *F*(4,116) = 2.16, *p* = 0.078). 

Compared to proposed cut-off scores by Ang et al. [8] based on the data of 479 healthy individuals, patient-reported apathy should be classified as moderate on at least one of the dimensions for three patients with AUD (14.3%), seven patients with ARCI (28.0%) and six patients with KS (24.0%) and as severe for six patients with AUD (28.6%) and five patients with KS (20.0%). Caregiver-reported apathy was classified as moderate on at least one of the dimensions for seven patients with AUD (35.0%), six patients with ARCI (30.0%) and eight patients with KS (36.4%) and as severe for three patients with AUD (15.0%), two patients with ARCI (10.0%) and five patients with KS (22.7%). The mean ratings, standard deviations and proposed cut-off scores by Ang et al. [8] are presented in Table A2 in Appendix A.

Table 1 shows the mean ratings and standard deviations of cognitive functioning assessed using the MoCA and mean ratings of everyday functioning assessed using the PCRS. Performance of the three patient groups differed on the MoCA total score (*F*(2,60) = 19.15, *p* < 0.001; AUD and ARCI > KS) and the MoCA memory index score (*F*(2,60) = 21.74, *p* < 0.001; AUD and ARCI > KS). Ratings of everyday functioning also differed across the three patient groups (*F*(2,67) = 13.54, *p* < 0.001; AUD and ARCI > KS). Table 2 shows the correlations between patient-reported and caregiver-reported apathy, everyday functioning and the MoCA total score. Significant correlations were present between caregiver-reported behavioral and social apathy and everyday functioning (*r* = −0.422, *p* = 0.001; *r* = −0.330, *p* = 0.011). Neither emotional apathy as reported by caregivers nor any dimension of patient-reported apathy correlated with everyday functioning. The correlations between patient-reported and caregiver-reported apathy and everyday functioning are also represented in Figure 2. Moreover, no significant correlations were found between patient-reported or caregiver-reported apathy and the MoCA total score.

Table 3 shows mean ratings and standard deviations of patient-reported and informant-reported apathy for patients with AUD, ARCI and KS for whom both versions of the AMI were completed. No differences between patient and caregiver ratings were present in the AUD group for behavioral, emotional or social apathy (*t*(17) = 0.61; *p* = 0.551; *t*(17) = −1.51, *p* = 0.150; *t*(17) = 0.08, *p* = 0.939). In the ARCI group, caregivers reported on average more severe behavioral and social apathy (*M* = 1.73, *SE* = 0.60; *M* = 1.96, *SE* = 0.54) compared to the patients themselves (*M* = 0.74, *SE* = 0.54; *M* = 1.36, *SE* = 0.66), *t*(19) = −6.13, *p* < 0.001, *d* = 0.72; *t*(19) = −4.20, *p* < 0.001, *d* = 0.64. Ratings of emotional apathy did not differ between patients and caregivers for the ARCI group (*t*(19) = −0.37, *p* = 0.713). In the KS group, caregivers reported on average more severe behavioral apathy (*M* = 1.92, *SE* = 0.82) compared to the patients themselves (*M* = 1.18, *SE* = 0.88; *t*(21) = −3.60, *p* = 0.002, *d* = 0.97). There were no differences between patient- and caregiver-ratings for emotional and social apathy in the KS group, *t*(21) = −0.82, *p* = 0.424; *t*(21) = −1.76, *p* = 0.093).

## 4. Discussion

This study was the first to examine the multidimensional nature of apathy in patients with AUD, ARCI and KS using the patient-report and informant version of the apathy motivation index (AMI). In addition, we examined differences between patient and caregiver ratings, as impaired awareness of deficits is present in patients with ARCI and KS. Furthermore, the relation between the three dimensions of apathy, everyday functioning and overall cognitive performance was investigated.

Overall, both patients and professional caregivers reported social apathy as more severe compared to behavioral and emotional apathy. Average apathy ratings did not differ between the three patient groups (AUD, ARCI and KS). Additionally, a higher level of social apathy was observed across all three patient groups compared to behavioral and emotional apathy. As hypothesized, discrepancies between patient and observer ratings were found within patients with ARCI and KS.

The finding that social apathy is more severe compared to behavioral and emotional apathy is different from previous results [16], showing more severe observer-rated cognitive apathy compared to emotional apathy using the apathy evaluation scale–informant (AES-I). However, it should be noticed that emotional apathy as measured by the AES is based on two items, while emotional apathy as measured by the AMI is based on six items and therefore covers a broader range of the total construct of apathy. Further the AES-I does not have a separate domain for social apathy. The current results therefore extend our previous findings [18].

In accordance with previous investigations of apathy in different patient samples (e.g., Parkinson’s disease (PD), Alzheimer’s dementia, autoimmune limbic encephalitis and subjective cognitive impairment) and healthy individuals [8,9,27], the current results corroborate the claim that social apathy is a separate dimension. In fact, for the current study sample social apathy turned out to be the most relevant dimension, consistently reported by both patients and caregivers. Future research should focus more on the relationship between social apathy and AUD. Social interactions might play an important role in the risk of relapse in excessive alcohol consumption, for example in the case of interaction with other users. However, social interactions might also have an important role in maintaining abstinence by giving support and combat loneliness.

Discrepancies between patient and observer ratings were found within patients with ARCI and KS. Formal caregiver of patients with KS reported more behavioral apathy compared to the patients themselves and caregivers of patients with ARCI reported more severe behavioral and social apathy compared to the patients themselves. Within the AUD group, there were no differences between patient and caregiver ratings. These findings support our hypothesis that patients with cognitive impairment may have limited insight into their own everyday functioning [18].

As hypothesized, caregiver-reported behavioral apathy was associated with everyday functioning. In addition, however, we also found caregiver-reported social apathy, but not emotional apathy, to be related to everyday functioning. An explanation for this is that both the behavioral and social apathy subscales include items that are clearly related to the patients’ activities, whereas the emotional subscale refers to the patient’s thoughts and feelings. In contrast to previous results [16] with the AES in KS patients, we did not find a relation between any of the AMI subscales and overall cognitive performance as measured with the MoCA. This may be understood in the context of the samples included in both studies. That is, the current study includes AUD, ARCI and KS patients in a psychiatric hospital setting, while the previous study only included KS patients who were residents of a specialized nursing home. Consequently, the latter group was more severely cognitively impaired and more apathetic than the current group. This may explain the strong association between these two symptom dimensions in the previous KS sample.

The current study expands existing literature on the prevalence of apathy. Compared to cut-off scores based on the data of healthy individuals [8], 42.9% of the patients with AUD, 28% of the patients with ARCI and 44% of the patients with KS can be classified as apathetic based on the patients’ self-reports. Based on the observer ratings, 50% of the patients with AUD, 40% of patients with ARCI and 59.1% of the patients with KS were classified as being apathetic on at least one of the dimensions. In a study with patients with PD, in which apathy is a characteristic symptom as well, 35.2% were classified with the AMI as being apathetic based on the same data of healthy individuals [30]. These prevalences of apathy in patients with alcohol use disorder with or without cognitive impairment are comparable or even higher than those in PD.

There is some evidence that the current findings can be understood in the context of dopaminergic dysfunction, which may play a role in apathy across different neuropsychiatric diseases including PD, schizophrenia and alcohol addiction [32]. However, it has also become apparent that dopamine depletion or dopamine receptor blockade is unlikely to be a complete explanation for the apathy syndrome. In patients with syndromes associated with frontotemporal lobe degeneration (FTLD) [33] and patients with PD who develop impulse control disorders on dopamine receptor agonists [34], apathy and impulsivity are commonly co-occurring and positively correlated. Moreover, there is also some evidence for a negative correlation between cortical thickness and severity of apathy in alcohol-dependent individuals, irrespective of cognitive impairment [13]. Future research might profitably focus on the relationship between apathy and impulsivity in AUD, ARCI and KS and its underlying mechanisms.

A strength of the current study is that the multidimensional nature of apathy is examined according to the recent literature in a broad study population including AUD, ARCI and KS. Furthermore, the results presented here include the view of patients themselves and show comparable ratings of emotional apathy between the patients and their professional caregivers. This suggest that despite limited insight into being behaviorally and socially apathetic, patients with KS and ARCI do have insight into their emotional sensitivity, again reinforcing the observation that there are dissociable domains of apathy.

A limitation of the current study is that depressive symptoms were not measured, while apathy is found to be overlapping with depression. Ang et al. [8] showed that the AMI correlates with Beck’s depression inventory (BDI), a measure for the severity of depressive symptoms. In the current study population, none of the patients met the criteria for major depressive disorder, substantiated by consultation of a psychiatrist. However, in future research on apathy, it is important to also assess depressive symptoms. In addition, the current results do not provide clarity on how the presence of apathy in individuals with alcohol addiction should be understood. It is yet unclear whether apathy is only the result of chronic and excessive alcohol consumption, affecting neural systems underlying apathy, as there is also evidence that apathy may be pre-existing in individuals with SUD [35]. More research on the course of apathy during addiction treatment is therefore needed. Furthermore, we only have the MoCA as global cognitive screening measure. The MoCA includes a valid memory index score, but does not include a valid subscore for executive dysfunction [29], making it not possible to examine the relation between the AMI subscales and executive function. In addition, as our sample sizes are relatively modest from a statistical perspective, replication of our results in larger samples is recommended.

## 5. Conclusions

The current results demonstrate that apathy is present in patients with AUD, ARCI and KS. In fact, severity levels of apathy and the multidimensional nature of apathy did not differ between these three patient groups. Both patients and caregivers evaluate social apathy as more severe compared to behavioral and emotional apathy. Furthermore, in accordance with previous literature, discrepancies in the evaluation of the severity of apathy between patients and their professional caregivers can be expected for patients with ARCI and KS. As apathy has been found to be a predictor of relapse in substance use disorder [36], our results emphasize the need for interventions aimed at reducing apathy in patients with alcohol use disorder with or without cognitive impairment. The current results also demonstrate the relevance of assessing apathy given its association with everyday functioning in individuals with alcohol addiction, irrespective of their cognitive impairments.

## Figures and Tables

**Figure 1 jcm-10-02447-f001:**
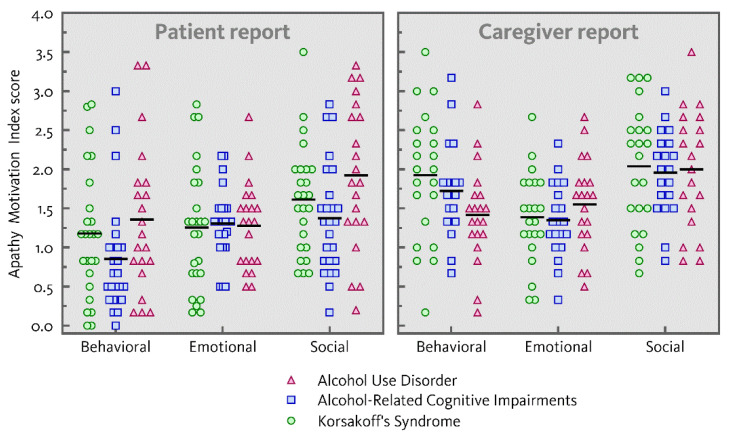
Individual mean ratings of behavioral, emotional and social apathy on the apathy motivation index in patients with alcohol use disorder, alcohol-related cognitive impairments and Korsakoff’s syndrome as reported by patients and caregivers.

**Figure 2 jcm-10-02447-f002:**
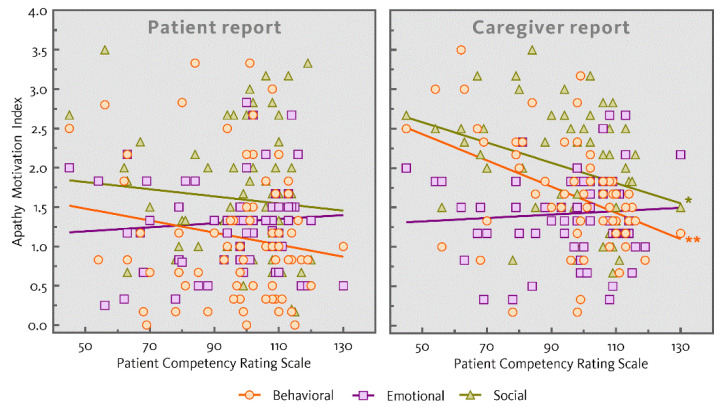
Scatterplots (with regression lines) showing the correlations between the patient and caregiver reports for the subscales of the apathy motivation index and the patient competency rating scale (* *p* < 0.05, ** *p* < 0.01).

**Table 1 jcm-10-02447-t001:** Group characteristics of patients with alcohol use disorder (AUD), alcohol-related cognitive impairments (ARCI) and Korsakoff’s syndrome (KS).

Characteristic		AUD		ARCI		*KS*		*p* Value
		(*N* = 23)		(*N* = 25)		(*N* = 25)		
	*N*	Mean (SD)	Range	Mean (SD)	Range	Mean (SD)	Range	
Age (years)	73	56.7 (9.3)	40–72	58.6 (8.3)	40–75	61.5 (6.7)	46–77	0.122
Sex distribution (men/women)	73	20/3		23/2		17/8		0.066
Abstinence duration (days) ^1^	73	48	38–171	51	39–1096	152	39–6992	0.002 ^7^
NART IQ ^2^	60	88.5 (14.5)	70–118	90.6 (14.1)	60–116	94.7 (16.1)	58–125	0.421
Full-scale WAIS-IV IQ ^3^	62	87.4 (15.2)	59–115	82.3 (12.1)	55–97	80.5 (13.5)	63–108	0.242
Educational level ^4^	73	4 (5)	1–6	5 (5)	2–7	5 (5)	2–7	0.321
MoCA ^5^ Total score	63	24.9 (3.3)	18–30	22.5 (2.9)	16–26	18.5 (3.9)	12–24	<0.001 ^8^
MoCA Memory Index Score	63	11.8 (3.3)	4–15	10.3 (3.0)	4–15	6.1 (2.3)	2–10	<0.001 ^8^
PCRS ^6^ Total Score	70	105.4 (10.2)	84–130	101.1 (11.9)	70–120	83.1 (21.6)	45–114	<0.001 ^9^

^1^ Median reported. ^2^ Estimated premorbid IQ score as measured with the Dutch version of the National Adult Reading Test [24]. ^3^ Means of full-scale IQ scores as measured with the Wechsler Adult Intelligence Scale, Fourth Edition (WAIS-IV) [25]. ^4^ Educational level (median and range) assessed using a Dutch classification range from 1 (less than primary school) to 7 (university degree). ^5^ Montreal Cognitive Assessment. ^6^ Patient Competency Rating Scale. ^7^ AUD < KS. ^8^ KS < ARCI and KS > AUD. ^9^ AUD and ARCI > KS.

**Table 2 jcm-10-02447-t002:** Pearson correlations between patient-reported and caregiver-reported behavioral, emotional, and social apathy assessed using the apathy motivation index, everyday functioning assessed using the patient competency rating scale (PSRS), and cognitive functioning assessed using the Montreal cognitive assessment (MoCA).

**Patient-Reported Apathy (*N* = 69)**	**PCRS Total Score**	**MoCA Total Score**
Behavioral apathy	−0.161	0.066
Emotional apathy	0.079	−0.063
Social apathy	−0.102	0.188
**Caregiver-Reported Apathy (*N* = 59)**	**PCRS Total Score**	**MoCA Total Score**
Behavioral apathy	−0.422 **	−0.119
Emotional apathy	0.069	0.167
Social apathy	−0.330 *	−0.179

* *p* < 0.05. ** *p* < 0.01.

**Table 3 jcm-10-02447-t003:** Mean ratings and standard deviations of patient-reported and caregiver-reported apathy assessed using the apathy motivation index for the patients with alcohol use disorder (AUD), alcohol-related cognitive impairments (ARCI) and Korsakoff’s syndrome (KS) for whom both versions were completed.

Level of Apathy	AUD		ARCI		KS	
	(*N* = 18)		(*N* = 20)		(*N* = 22)	
	Patient-Reported	Informant-Reported	Patient-Reported	Informant-Reported	Patient-Reported	Informant-Reported
Behavioral apathy	1.44 (0.97)	1.34 (0.60)	0.74 (0.54)	1.73 (0.60)	1.18 (0.88)	1.92 (0.82)
Emotional apathy	1.28 (0.60)	1.56 (0.61)	1.30 (0.45)	1.35 (0.48)	1.21 (0.82)	1.39 (0.58)
Social apathy	1.98 (0.91)	1.96 (0.79)	1.36 (0.66)	1.96 (0.54)	1.64 (0.71)	2.04 (0.79)

## Data Availability

Data and the Apathy Motivation Index are available from the authors upon request.

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
