# Peer review of "Behavioral, Emotional and Social Apathy in Alcohol-Related Cognitive Disorders"

_jcm, 2021, doi:10.3390/jcm10112447_

Round 1
Reviewer 1 Report
The authors state that the aim of the study was “to examine the multidimensional nature of apathy in patients with KS, ALC, and AUD.” Apathy ratings (assessed with the Apathy Motivational Index – AMI) were obtained from both participants and their professional caregivers and these scores were related to everyday functioning (assessed by the Patient Competency Rating Scale- PCRS) and overall cognition (assessed by the MoCA). Participants included 24 individuals with KS, 24 individuals with alcohol-related cognitive impairments (ALC), and 21 individuals with alcohol use disorder without cognitive impairments (AUD). The AMI assesses three types of apathy – behavioral, emotional, and social apathy. Groups did not differ on apathy rating. Social apathy was the most prominent type of apathy reported by both patients and caregivers. Caregivers reported relations between behavioral and social, but not emotional, apathy and everyday functioning. The authors concluded that “these results show that apathy is present in a substantial proportion of patients with alcohol addiction with or without cognitive impairments.”
Knowledge of the prevalence and effect of apathy in the context of alcohol use disorders is a clinical relevant issue, one that may also provide information as to underlying brain-behavior relationships. Although this is an interesting topic of investigation, this study has a number of limitations- some of which the authors noted in their discussion section - the most important in my opinion is the omission of a depression symptom measure such as the BDI-II. Overall, the authors have to provide more information and make clearer the limitations as to the independent and dependent measures used to assess the constructs reported (apathy, everyday functioning, and general overall cognition). In addition, scatterplots depicting significant correlations (to ensure that outliers are not driving the relationship) and more transparency as to their methodological decisions are warranted.
Other points to be addressed:
The acronyms ALC and AUD are confusing. If ALC is to reflect alcohol-related cognitive impairment maybe use ARCI or AUD+CI?
The authors do not have any evidence that the patients within their AUD group have recovered fully – these participants may never have demonstrated cognitive impairment. Please revise the statement starting on line 35.
What was the reasoning behind the decision to separate the participants with AUD with and without cognitive impairment right from the beginning?
Speculation as to the neural correlates of apathy would be a worthwhile addition to the manuscript – relation to executive dysfunction?
Is apathy a precursor/risk factor of alcohol addiction or a consequence – what are the authors thoughts?
What is the reliability and validity of the AMI in this population? The majority of the original population that the AMI was standardized on were students and full-time employed adults (Ang et al., 2017). Some of the questions do not seem particularly relevant to a KS population. What was the reasoning behind choosing the AMI to assess apathy in this population?
Can the authors provide more information about the PCRS? Why was this specific measure chosen?
Do the authors have estimates of premorbid IQ for participants?
More information is needed about participants – how much alcohol consumed? Length of disease? Length of abstinence (more than stating that participants were at least 6 weeks into abstinence before testing). The reader does not get a good feel of the participants included in the study.
How were “disorders” defined? Were scores to be -1SD, -2SD from age- and education-matched healthy controls? How was cognitive impairment defined to distinguish the AUD from the ALC subgroups in this study? It is a limitation that the authors did not have their own control group.
Table 1: Include range – minimum and maximum –scores for all continuous variables.
The authors may want to consider using total scores as opposed to average score when reporting apathy subscale scores- it is easier to see the range of scores.
Table 2: Suggest that ** denote greater level of significance p<.01 and * p<.05.
Line 209: State the AMI cutoff scores.
Scatterplots depicting relations between PCRS Total Score and apathy scores would be useful to demonstrate to the reader that outliers did not drive the correlation.
What are the correlations among the different apathy subscales (behavioral activation, emotional sensitivity, and social motivation) for each group?
Can the authors cite published research supporting the notion that reducing apathy reduces relapse rate of substance use disorders? This is stated without reference.
Although the MoCA assesses general cognitive level – this should be made clear that this is a screening measure and that the lack of findings related to this score may not truly reflect the relation between apathy and cognitive functioning – particularly executive function abilities.
Minor editorial changes:
Line 260: “day” should be added to “every” before the word “functioning”
Line 276: Specify which populations the authors are referring to
Line 277: Typo “that that”
Reviewer 2 Report
In the paper Behavioral, emotional and social apathy in alcohol-related cognitive disorders the authors wish to relate three dimensions of apathy to everyday functioning and overall cognitive performance of individuals, suffering from AUD, AUD in addition to cognitive disorders and individuals, suffering from Korsakoff’s syndrome. The authors hypothesize that all three dimensions of apathy will be present in all patient groups. They expect behavioral apathy to be the most frequently reported dimension of apathy in all groups and hypothesize that especially behavioral apathy is negatively related to everyday functioning and cognitive performance. The authors also hypothesize that professional caregivers will report higher levels of apathy among the patients than patients with KS and ALC themselves.
The paper is well-written and the topic of interest. I do miss, however, a more thorough description of the patients involved in the study, in particular it should be described how long the patients have been abstinent and whether the mean period of abstinence differs between the groups. Currently, it is just described that all patients have been abstinent for at least 6 weeks. I thus wonder if (in particular) there are significant differences between the groups in how long they have been abstinent, and if so, if it should be taken into consideration in the analysis of differences in scores between the groups.
I also wonder if it is really true that severity levels of apathy and the multidimensional nature of apathy did not differ between the three patient groups. Does the study have sufficient power to allow for such firm conclusion? Furthermore, it is stated in the methods section that the AUD-group includes individuals with severe AUD only (and thus not all levels of AUD) and that, at least, should be noted in the conclusion.
Minor comments:
N-values should be added to table 2.
Round 2
Reviewer 1 Report
The authors have been responsive to almost all of my points of concern; however, 2 points of concern remain:
(1) It is still unclear as to how cognitive impairment was determined. The authors state, “The neuropsychological assessment covered the domains of attention, memory, executive functioning, visuospatial functions, and social cognition, and scores were interpreted by experienced clinical neuropsychologists using all available age-, education, and gender-adjusted Dutch normative data and/or available cut-off scores.” Was “impairment” defined as scores that were 1 sd below the mean for normative group? 2 sd below the mean?
(2) The range of estimated premorbid IQ was broad and participants below estimated IQ of 70 were included in the ARCI and KS groups. Why include participants with estimated premorbid IQ < 70? How many participants in the ARCI and KS groups had estimated IQ below 70 and did the results remain unchanged when those participants were excluded?
